# TERT Promoter Revertant Mutation Inhibits Melanoma Growth through Intrinsic Apoptosis

**DOI:** 10.3390/biology11010141

**Published:** 2022-01-14

**Authors:** Yanbing Wang, Yiwu Chen, Chang Li, Zhiwei Xiao, Hongming Yuan, Yuanzhu Zhang, Daxin Pang, Xiaochun Tang, Mengjing Li, Hongsheng Ouyang

**Affiliations:** 1Key Laboratory for Zoonoses Research, Ministry of Education, College of Animal Sciences, Jilin University, Changchun 130062, China; wangyb19@mails.jlu.edu.cn (Y.W.); chenyw20@mails.jlu.edu.cn (Y.C.); xiaozw9918@mails.jlu.edu.cn (Z.X.); yuanhongming@jlu.edu.cn (H.Y.); yuanzhu16@mails.jlu.edu.cn (Y.Z.); pdx@jlu.edu.cn (D.P.); xiaochuntang@jlu.edu.cn (X.T.); 2College of Plant Sciences, Jilin University, Changchun 130062, China; lichang8220@mails.jlu.edu.cn; 3Chongqing Research Institute, Jilin University, Chongqing 401123, China; 4Chongqing Jitang Biotechnology Research Institute, Chongqing 401123, China

**Keywords:** TERT promoter, revertant mutation, melanoma, Bcl-2, intrinsic apoptosis

## Abstract

**Simple Summary:**

TERT -146 C>T frequently occurs in many cancer cells. Research targeting the telomerase reverse transcriptase (TERT) promoter contributes to a better understanding of cancer development and treatment. Many conventional cancer treatments aim to develop new drugs targeting TERT. Here, for TERT -146 we converted T to C. The proliferation, migration and invasion of melanoma cells in vitro, and the growth of the tumor in vivo were inhibited. Moreover, the downregulated protein expression of B-cell lymphoma 2 (Bcl-2) indicated that the TERT promoter revertant mutation abrogated the inhibitory effect of mutant TERT on apoptosis. These data elucidated the relationship between the TERT promoter revertant mutations and apoptosis for the first time, and also implied that TERT -146 may be a causal mutation of melanoma. This study provides a new insight into the TERT promoter revertant mutations and apoptosis. The TERT promoter provides preliminary validation of the potential tumor treatment.

**Abstract:**

Human telomerase is a specialized DNA polymerase whose catalytic core includes both TERT and human telomerase RNA (hTR). Telomerase in humans, which is silent in most somatic cells, is activated to maintain the telomere length (TEL) in various types of cancer cells, including melanoma. In the vast majority of tumor cells, the TERT promoter is mutated to promote proliferation and inhibit apoptosis. Here, we exploited NG-ABEmax to revert TERT -146 T to -146 C in melanoma, and successfully obtained TERT promoter revertant mutant cells. These TERT revertant mutant cells exhibited significant growth inhibition both in vitro and in vivo. Moreover, A375^−146C/C^ cells exhibited telomere shortening and the downregulation of TERT at both the transcription and protein levels, and migration and invasion were inhibited. In addition, TERT promoter revertant mutation abrogated the inhibitory effect of mutant TERT on apoptosis via B-cell lymphoma 2 (Bcl-2), ultimately leading to cell death. Collectively, the results of our work demonstrate that reverting mutations in the TERT promoter is a potential therapeutic option for melanoma.

## 1. Introduction

Melanocytic carcinoma are the most aggressive type of skin cancer, ranging from benign lesions, known as congenital melanocytic nevus, to malignant lesions known as melanomas [1]. In 2019, the incidence of melanoma was 0.00144% (male: 0.00077%; female: 0.00067%) on a global scale; therefore, 0.23% (male: 0.12%; female: 0.11%) of patients will die (http://www.healthdata.org/ (accessed on 1 September 2021)) [2,3]. Melanoma is caused by multiple factors, including chronic sun-damaged (CSD), aging, and gene mutations [4]. In particular, somatic mutations occur in dominant melanoma oncogenes during melanoma progression, such as *BRAF* (>50%) [5,6,7], *p16^INK4A^* (40%) [8,9], *NRAS* (21%) [5], *TERT* (77%), *PTEN*, *TP53*, and *ARID2*, etc. Such point mutations drive the progression from benign intermediate lesions to malignant melanoma [4]. Although most of the functions of these mutations in cancer progression have been verified, there is still a large gap in research on the use of these point mutations for cancer therapy.

TERT is a specialized reverse transcriptase responsible for the addition of multiple telomeric hexamer TTAGGG repeats to the ends of chromosomes to maintain chromosome integrity [10]. Point mutations in the TERT promoter range are long known to cause telomerase activation and cell proliferation of all types of melanoma cells [11,12]. Located at two hotspot positions on Chr5, 1,295,228 C>T (45% of all TERT mutations) and 1,295,250 C>T (15% of all TERT mutations) (hereafter termed -124 C>T and -146 C>T), these mutations are upstream of the transcription start site [12,13]. These two mutations are functionally exclusive, and they both create an identical 11 bp binding motif “CCCGGAAGGGG”, which is capable of binding to the ETS family of transcription factors, including the multimeric GA-binding protein A (GABPA), GA-binding protein B (GABPB), and ETS1 [14,15]. This causes a two- to four-fold increase in the transcriptional activity of mutant TERT [16,17], and the upregulation of ETS, GABPA, and GABPB1 proteins promotes the expression of mutant TERT [15], implying that the downregulation of GABPA and GABPB1 protein expression may affect the expression of TERT. Considering the decisive role of these mutations, TERT has been considered a promising target for drug development; therefore, studying TERT mutations is important for cancer prevention, screening, early diagnosis, and targeted gene therapy [18].

β-Catenin is an essential component of the Wnt/β-Catenin signaling, forming a complex with the ternary complex factors (TCFs) family of transcription factor members in the nucleus to regulate the transcription of target genes. Dysregulation of this signaling is frequently observed in cancers. TERT is directly regulated by Wnt/β-Catenin signaling [19,20]. It is not clear whether the TERT promoter revertant mutation affects Wnt/β-Catenin signaling.

Previous reports have demonstrated that apoptosis is inhibited by mutant TERT in tumor cells [21]. Moreover, mutant TERT can promote cell proliferation and invasion [22]. Inhibitory drugs targeting TERT, and the knockdown of TERT by siRNA or shRNA, can abrogate the inhibitory effect of TERT on apoptosis to induce cell death [21,22,23,24]. Members of the Bcl-2 family play a central role in the regulation of intrinsic apoptosis by modulating the expression of pro- and anti-apoptotic proteins [25]. As mentioned previously, when apoptosis occurs, the expression of anti-apoptotic proteins (e.g., Bcl-2) is downregulated, while the expression of pro-apoptotic proteins is upregulated, resulting in large amounts of cytochrome C (Cyt C) being released from the mitochondria to the cytoplasm, which leads to the activation of caspase 9 and, ultimately, intrinsic apoptosis [26,27,28]. It has been reported that converting TERT -124 T to C using CjABE causes glioblastoma (GBM) cell senescence and inhibits GBM growth in vitro and in vivo [11], and knockout of the TERT coding region can inhibit cancer cell proliferation [29]. Whether TERT promoter revertant mutations are associated with cell senescence, proliferation, or even apoptosis remains unclear.

Base editing (BE) systems have recently been developed that efficiently induce single nucleotide changes without forming double-stranded DNA breaks (DSBs). Currently, DNA base editors can be categorized as adenine base editors (ABEs) and cytosine base editors (CBEs). ABEs have been used to convert A/T to G/C, and CBEs have been used to convert C/G to T/A [30]. Recently, a novel ABE for an NG protospacer-adjacent motif (PAM) named NG-ABEmax has been reported that may be a promising method for gene disruption and editing that can be applied in regenerative medicine, biomedical research, medical oncology, and gene therapy [31,32,33,34,35,36,37,38,39]. 

Here, we hypothesized that TERT promoter revertant mutations would inhibit melanoma growth. In the present study, we converted -146 T to C using NG-ABEmax and obtained TERT promoter revertant mutant cells A375^−146T/C^ and A375^−146C/C^. As expected, the TERT promoter revertant mutation inhibited melanoma growth in vivo and in vitro through intrinsic apoptosis. Our study provides a novel potential option for melanoma treatment. Importantly, this is, to the best of our knowledge, the first study to reveal a relationship between TERT promoter revertant mutation and intrinsic apoptosis. 

## 2. Materials and Methods

### 2.1. Materials

Rabbit anti-TERT polyclonal antibody (#abs136649; 1:1500) was obtained from Absin. Rabbit anti-ETS1 polyclonal antibody (#A00931; 1:1500); mouse anti-β-Actin antibody (BM5422; 1:1500); mouse anti-β-tubulin antibody (#BM3877; 1:1500); rabbit polyclonal anti-PTEN antibody (#PB0423; 1:1500); rabbit polyclonal anti-Catenin-β antibody (#BM3905; 1:1500); mouse anti-c-MYC antibody (#BM0238; 1:1500); rabbit anti-BLC2 monoclonal antibody (#BM4985; 1:1500); rabbit anti-BAX monoclonal antibody (#BM3964; 1:1500); and rabbit polyclonal anti-VDAC1 antibody (#BA3754; 1:1500) were obtained from Boster Biological Technology. Rabbit anti-GABPA polyclonal antibody (#21542-1-AP; 1:1500); rabbit anti-GABPB1 polyclonal antibody (#12597-1-AP; 1:1500); rabbit caspase 9 polyclonal antibody (#10380-1-AP; 1:1500); rabbit ANT1/2 polyclonal antibody (#17796-1-AP; 1:1500); and rabbit cytochrome C polyclonal antibody (#10993-1-AP; 1:1500) were obtained from Proteintech. AIF rabbit monoclonal antibody (#AF1273; 1:1500) was obtained from Beyotime.

### 2.2. Cell Lines and Cell Culture Conditions

The human A375 cell line (a gift from the Department of Translational Medicine, The First Hospital of Jilin University, Changchun 130061, China), A375^−146T/C^ and A375^−146C/C^, were maintained in RPMI 1640 (obtained from the American Type Culture Collection) medium supplemented with 10% fetal bovine serum (PAN). The human HepG2 cell line (ATCC) and 293 cell line was maintained in DMEM (Gibco), supplemented with 10% fetal bovine serum (PAN).

### 2.3. Plasmid Construction

The ABEmax-NG plasmid was obtained from the Key Laboratory of Zoonosis Research, Ministry of Education, College of Animal Science, Jilin University, Changchun, China. To construct the vector expressing the sgRNA and SpCas9-NG, annealed and phosphorylated sgRNA backbone oligonucleotides (forward, 5′-GACCCGGAAGGGGTCGGGACG-3′; and reverse, 5′-CGTCCCGACCCCTTCCGGGTC-3′) were ligated into the plasmid digested with Spe I (#R0133S; NBE).

### 2.4. Electroporation, Selection of A375 and PCR

Cells were cultured in 10 cm dishes. Then, 10 µg of plasmids were transfected into 2.5 × 10^6^ A375 cells using a Neon Transfection System (MPK5000S; Invitrogen, Carlsbad, CA, USA), whereby the adherent cell fusion rate reached 80%. Cells were seeded into 10 cm plates (1500–2000 per well) for approximately 8 days. When single cell colonies formed, the selected cells were seeded in 24-well plates (1 clone per well) and cultured with RPMI 1640 (Gibco, Grand Island, NY, USA) medium supplemented with 20% FBS (Gibco, Grand Island, NY, USA). Once cell clones reached 80%, clones were digested with 0.25 trypsin, and 20% of the single clones were lysed in 10 µL of NP40 buffer (1% Proteinase K, 1% NP40, 10% Standard Taq Reaction Buffer, 88% ddH_2_O). The samples were incubated at 56 °C for 60 min and then at 95 °C for 10 min to inactivate proteinase K and facilitate DNA denaturation. PCR amplification was performed as previously [11]. The PCR products were separated by electrophoresis. Sanger sequencing was performed to assess the DNA sequence of the PCR products.

### 2.5. Real-Time Q-TRAP

As previously described [40,41], in short, cells were lysed with precooled NP-40 lysis buffer, the supernatant was collected after centrifugation, and the total protein concentration of the samples was determined using the BCA protein assay kit. Total protein from 293 cells (125 ng, 250 ng, 500 ng, 750 ng, and 1000 ng) was used to make the standard curves. A quantity of 500 ng total protein of A375^−146T/T^, A375^−146T/C^, and A375^−146C/C^ was used to assay. Q-TRAP reaction system: RNase/DNase-free H_2_O, 2.8-x µL; SYBR Green PCR master mix (Tiangen, Beijing, China), 5 µL; EGTA (1 mM), 1 µL; ACX primer (10 µM), 0.6 µL; TS primer (10 µM), 0.6 µL; sample x µL. The mixture is incubated in the dark at 22–30 °C for 30 min to extend the substrate by telomerase. Then, 95 °C for 15 min, 40 cycles at 95 °C for 10 s, and at 60 °C for 32 s. Produced standard curves and calculated RTA units for samples are to be tested. Normalized the RTA to that of A375^−146T/T^. The RTA was quantified as described in the method. For the same Q-TRAP, the RTA values of the unknown samples were calculated using the following equation: y = −0.0142x + 1.308, R^2^ = 0.9611; RTA = 10 ^[(Ct sample − Y int)/slope]^.

### 2.6. Telomere Length Measurements

Telomere length measurements were performed as described previously [42]. Genomic DNA was extracted from A375^−146T/T^, A375^−146T/C^, A375^−146C/C^, and HepG2. A quantity of 0.5 ng extracted DNA was used to be template of quantitative PCR. Telomere length: ΔCt = Ct^tel^ − Ct^HGB^; ΔCt1: A375^−146T/T^, A375^−146T/C^, or A375^−146C/C^; ΔCt2: HepG2; and T/S = 2^−(ΔCt1−ΔCt2)^. The primers used and detailed method were as described previously [43].

### 2.7. Transwell Migration and Invasion Assays

Cell migration and invasion were detected by using Transwell assays (Corning). For the cell migration assay, 2000 A375^−146T/T^ and A375^−146T/C^ or A375^−146C/C^ cells were seeded in the upper nested chambers, and the chambers were then placed into 24-well plates with 600 μL of medium containing 10% FBS in each well. Twenty-four hours later, the medium on the lower surfaces was discarded, the lower surfaces were washed twice with PBS for 3 min each time, and then a fixed solution containing 4% paraformaldehyde was added. The cells were then washed twice with PBS for 3 min each time. Finally, the cells were stained with crystal violet or DAPI. For the invasion assay, the upper nested Transwell chambers were precoated with 50 µL Matrigel diluted with RPMI 1640 (1:8 dilution). The remaining steps were the same as those for the cell migration assay. Images were taken by an inverted microscope (Olympus) and repeated three times. Each experimental replicate followed the same approach, and the measurements were repeated three times. Images were analyzed and cells counted by ImageJ.

### 2.8. Off-Target Analysis

For off-target analysis in edited cell lines, potential off-target sites were selected using Cas-OFFinder (http://www.rgenome.ne-146T/Cas-offinder (accessed on 1 September 2021)). The amplified products were sequenced by Sanger sequencing. The primers used are listed in Figure 3A.

### 2.9. Reverse Transcriptase-PCR

Total RNA was extracted from the TERT promoter revertant mutation cells, and 2 μg Total RNA was used to synthesize complementary DNA (#KR103; Tiangen Biotech, Beijing, China). Then, 1 µL of cDNA was added to a 10 μL PCR mixture. SuperReal PreMix Plus (#FP205; Tiangen Biotech, Beijing, China) was used to determine the threshold-cycle value for each sample using a q225 real-time PCR detection system (Novogene, Beijing, China). Duplicate real-time PCR analyses were performed for each sample, and the obtained threshold-cycle (CT) values were averaged. In these studies, samples were normalized to β-actin. The relative expression levels for the target genes were determined using the fold change (2^−ΔCt^) method (ΔCt = the threshold cycle for β-actin—the threshold cycle for the target gene). The sequences of the PCR primers used were previously described [11].

### 2.10. Western Blotting

Cells were lysed in culture total protein extraction reagent (#AR0103; Boster, San Francisco, CA, USA) containing 1/100 PMSF. Briefly, lytic products were boiled at 95 °C for 5 min to denature and separated by sodium dodecyl sulfate polyacrylamide gel electrophoresis. Then, 40 µg of protein was transferred to PVDF membrane (#AR0136; Boster, San Francisco, CA, USA) and sealed with 5% skim milk in TBST. The membranes were then sectioned according to the molecular weight, and the membranes were incubated with primary antibodies at 4 °C for 12 h, washed with TBST three times, and incubated with horseradish peroxidase-conjugated secondary antibodies at room temperature for 1.5 h. Western blot signals were detected by ultrasensitive ECL chemiluminescence ready-to-use substrate (#AR1196; Boster, San Francisco, CA, USA).

### 2.11. Cell Proliferation Assays

For the cell proliferation assay, 2000 cells were seeded on 96-well plates, and every 12 h the original medium was discarded. The cells were washed once with PBS, and then 100 µL of culture medium and 10 µL of CCK 8 solution were added. The CD value of each well was measured after 1 h, and the number of cells per well was calculated. At least three replicates per sample were analyzed.

### 2.12. Tumor Xenografts

After digestion with 0.25% trypsin containing EDTA, the cells were washed twice with precooled PBS. Cells were resuspended in 0.1 mL Matrigel (#356234; Corning, Corning, New York State, USA), and the suspension was injected subcutaneously into the anterior ventral region of each nude mouse (Charles River; Beijing, China). The nude mice were divided into three groups, with a total of four mice in group a, in which 4 × 10^6^ A375^−146C/C^ was injected into the left anterior ventral side of each nude mouse, and the same amount of A375^−146T/T^ was injected into the right anterior ventral side; six nude mice were included in group b and c, in which 2.5 × 10^6^ A375^−146T/T^ or A375^−146C/C^ were injected into the right or left anterior ventral side only. After the injection, size of each tumor was measured every two days by a Vernier caliper, and tumor volume was calculated as volume = L × W^2^ × Π /6 with L and W representing the largest and smallest diameters, respectively. The mice were euthanized when L > 1 cm and the volume did not exceed 2000 mm^3^.

### 2.13. DNA Damage Assay (γ-H2AX Immunofluorescence Assay)

Cells were seeded into 12-well cell culture plates as described in the instructions, and after 12 h, cells were washed once with PBS, fixed for 10 min by adding fixative, and washed 3 times for 5 min each time with washing solution. Immunostaining closure solution was added and closed at room temperature for 10–20 min. The immunostaining closure solution was aspirated, γ-H2AX rabbit monoclonal antibody was added and incubated for 1 h at room temperature. It was washed 3 times with washing solution for 10 min each. Anti-rabbit 488 was added and incubated for 1 h at room temperature. It was washed 2 times with washing solution for 10 min each time. Cell nuclear staining solution (DAPI) was added and it is stained for about 5 min at room temperature. The nucleus staining solution was aspirated and washed 3 times with washing solution for 5 min each time. It was then observed with a fluorescence microscope and photographed.

### 2.14. Statistics

Two-tailed Student’s *t*-test was used to perform statistical analysis. All *p* values (Student’s *t*-test) were two-sided, and *p* < 0.05 was considered to indicate significance. All analyses were performed using GraphPad Prism (version 8.0.2.263 for Windows; GraphPad Software, Inc. San Diego, CA, USA).

## 3. Results

### 3.1. TERT Promoter Revertant Mutation Inhibits TERT Expression

It is well established that overexpression of TERT is associated with the ETS binding motif and the expression of ETS family members [15,17,44]. TERT -146 C>T activates telomerase and enables cancer cells to achieve unlimited proliferation [13]. To explore the effect of TERT -146 T>C on melanoma cells, we used the NG-ABEmax to edit TERT -146 and screened heterozygotes and homozygotes, as shown in the Sanger sequencing results. (Figure 1A). The mutant TERT promoter created a new binding motif for ETS transcription factors and TCFs near the transcription start, and then ETS family members bind with mutant TERT promoter, resulting in the activation of TERT transcription [17] (Figure 1B). The -146 T>C decreased the ETS binding motif (Figure 1A). The TERT promoter revertant mutation abrogated the binding motif generated by -146 C>T [11,42]. Next, we measured telomerase activity, as previously described [40,41]. Briefly, whole proteins were extracted from the cells with NP-40 lysate, and then the protein concentration of the samples was determined using the BCA protein concentration assay kit, and 500 ng of protein was taken for the real-time Q-TRAP. The results showed that TERT revertant mutant cells had significantly lower telomerase activity compared to A375, indirectly reflecting that the TERT revertant mutation inhibits TERT promoter activity (Figure 1C). Then, we extracted RNA and protein from the cells and detected the expression level of TERT by RT-qPCR and Western blot. As expected, TERT transcription and protein expression were downregulated in A375^−146T/C^ and A375^−146C/C^ cells; in particular, TERT was barely expressed in A375^−146C/C^ cells (Figure 1D). The Western blot results showed that the expression of ETS1, GABPA, and GABPB1 was downregulated in A375^−^^1^^46T/C^ and A375^−146C/C^ cells (Figure 1E–G and Appendix A). The downregulation of GABPA and GABPB1 protein expression reduced the GABPA/GABPB1 complex, thereby downregulating the expression of TERT [15,44], and TERT expression was also suppressed by the downregulation of ETS1 (Figure 1B) [17,44]. 

### 3.2. TERT Promoter Revertant Mutation Reduces Telomere Length, Migration and Invasion, and Induces Cellular Deformation

We investigated whether TERT promoter revertant mutation affected the telomere length and proliferation of cells. The TERT promoter was mutated when the cells became cancerous (-146 C>T), and this mutation led to apoptosis inhibition and unlimited proliferation [21]. The TERT promoter revertant mutation could inhibit proliferation of malignant glioma cells in vitro [11]; however, at the same time, we hypothesized that the TERT revertant mutation can induce growth inhibition and apoptosis (Figure 2A). Quantitative PCR (q-PCR) was used to determine the telomere lengths in A375^−146T/T^, A375^−146T/C^, and A375^−146C/C^ cells as described in the method. The results showed shortened telomere lengths for both A375^−146T/C^ and A375^−146C/C^ compared with A375^−146T/T^, with A375^−146C/C^ shortening more than A375^−146T/C^ (Figure 2B). For the migration assay, we then measured the migration and invasion by Transwell. A total of 2000 A375^−146T/T^ and A375^−146T/C^ or A375^−146C/C^ cells were seeded in the upper nested chambers, and the chambers were then placed into 24-well plates with 600 μL of medium containing 10% FBS in each well. For the invasion assay, the upper nested Transwell chambers were precoated with 50 µL Matrigel diluted with RPMI 1640 (1:8 dilution). We observed and recorded the number of cells in the chambers below after 20 h. Migration and invasion were also significantly inhibited in A375^−146T/C^ and A375^−146C/C^, especially in A375^−146C/C^, with little migration or invasion detected (Figure 2C,D). Cell morphology was observed microscopically, and the morphology of A375^−146T/C^ and A375^−146C/C^ changed compared with A375^−146T/T^, including cellular shrinkage (triangle markers), the condensation and margination of chromatin (pentagon markers), and ruffling of the plasma membrane (arrow markers) (Figure 2E). Cells were cultured in 12-well plates and cells were sent to 12-well plates for 12 h and then fixed with fixative solution, and labeled with an anti-γ-H2AX antibody with green fluorescent. The results showed that both TERT revertant mutant cells had DNA damage compared to wild-type cells. These results indicated that A375^−146T/C^ and A375^−146C/C^ cells underwent apoptosis [45]. In previous studies, the revertant mutation of TERT -124 in glioma cells induced cellular senescence [11]; however, we found that senescence-associated biomarker β-galactosidase activity was inconspicuous in A375^−146T/C^ and A375^−146C/C^ cells. These results suggested that revertant mutation in TERT -146 of melanoma led to altered cell shape, reduced migration and invasion, damage and death by inducing apoptosis.

### 3.3. Off-Target Analysis

Many studies have shown that off-target effects are not only related to sgRNA but also to experimental conditions [46]. Off-target effects have been an important issue hindering the application of gene editing technology [47]. To detect off-target mutations in potential off-target sites, twelve potential off-target sites were screened and primers were designed for each off-target site (Figure 3A). DNA was extracted from A375^−146C/C^ and sequenced using Sanger sequencing (Figure 3B). The results showed that no off-target mutations were detected in the predicted 12 potential off-target sites. This means that the targeting of NG-ABEmax is relatively precise.

### 3.4. TERT Promoter Revertant Mutation Inhibits Melanoma Growth In Vitro and In Vivo

First, we examined the growth of A375^−146T/T^, A375^−146T/C^, and A375^−146C/C^ cells in vitro. Cells were seeded in 96-well plates and the number of cells per well was measured every 12 h using the CCK8 kit. The results showed that the proliferation of A375^−146C/C^ was decreased, and the cells showed slow growth during the 48 h in culture. After 48 h, the cells hardly proliferated, suggesting that the TERT promoter revertant mutation can inhibit the growth of melanoma in vitro (Figure 4A). A375^−146T/T^ and A375^−146C/C^ were injected into the anterior ventral region of 6- to 7-week-old nude mice to assess the growth of the cells in vivo (Figure 4B). The same number of A375^−146C/C^ cells and A375^−146T/T^ cells were injected into both sides of nude mice in group a (right: A375^−146T/T^, left: A375^−146C/C^) (Figure 4C). To prevent the interaction of tumors from both sides of the mice, A375^−146C/C^ cells or A375^−146T/T^ cells were only injected into the right anterior ventral region of 6- to 7-week-old nude mice in group b or group c, respectively (group b: A375^−146T/T^, group c: A375^−146C/C^) (Figure 4D). The results from groups a, b, and c showed that A375^−146T/T^ rapidly formed large tumors under the skin of nude mice, while A375^−146C/C^ cells were able to form tumors or formed only very small tumors under the skin of nude mice, suggesting that the TERT promoter revertant mutation can inhibit the growth of melanoma in vivo (Figure 4E). H&E staining of typical tumor tissue sections (group a, No. 4) showed that A375^−146T/T^ cells (No. 4 right, right image) formed a dense, radial tumor with a large distribution of blood vessels, and had a large number of epithelioid melanoma cells in the process of division (triangle marker). These tumors showed a high growth capacity and aggressiveness. Tumors formed by A375^−146C/C^ (No. 4 left, right image) were loosely structured (triangle marker), contained many cavities (pentagram marker), most of the cells were irregular, had less cytoplasm, and the cells proliferated slowly or even not at all (Figure 4F). Overall, TERT promoter revertant mutation not only inhibited the growth of melanoma in vitro, but also in vivo.

### 3.5. TERT Promoter Revertant Mutation and Wnt/β-Catenin Signaling

Wnt/β-Catenin signaling is a classic signaling pathway in cancer cells. *β-catenin* is a key effector of the Wnt/β-catenin signaling [48]. It has been shown earlier that telomeres are shorter in the mouse embryonic stem cells with deficient β-catenin and longer in the β-catenin-activated cells [20]. The knockout of PTEN upregulates β-catenin expression [49]. C-MYC is a cell cycle regulator located downstream of Wnt/β-Catenin, [50] β-catenin accumulation in the nucleus can upregulate c-MYC, and increased c-MYC induces proliferation of cancer cells [51]. We want to know whether the TERT promoter revertant mutation inhibits growth of melanoma via the Wnt/β-Catenin signaling. The results of the Western blot and RT-PCR showed that β-catenin and c-MYC transcription and protein expression, and protein expression of PTEN, were not significantly changed in A375^−14^^6T/C^ and A375^−14^^6C/C^ cells compared with A375^−146T/T^ cells (Figure 5A–C) (*p* > 0.05). We speculated that that the TERT promoter revertant mutations have no significant effect on Wnt/β-Catenin signaling.

### 3.6. TERT Promoter Revertant Mutation Activates the Intrinsic Apoptosis Pathway

The intrinsic apoptosis pathway has been shown to bring about the release of cytochrome C from the mitochondria into the cytosol, which, conclusively, led to the activation of caspase 9 and, in turn, to cell death by impairment of the transmembrane potential in the mitochondria [25,52]. Previous observations on cell morphology implied that the revertant mutant cells might have undergone apoptosis. We then performed FACs analysis with an Annexin V-FITC/PI Apoptosis Kit according to the instructions. We found that early apoptosis and late apoptosis were detected in A375^−146C/C^, and only late apoptosis was detected in A375^−146T/C^ cells. These results suggest that TERT revertant mutant cells do undergo apoptosis (Figure 6A–C). To further study, we examined the expression of intrinsic apoptosis-related proteins. The proteins of the cells were extracted with NP-40 lysate and subsequently subjected to Western blot. The results showed that the TERT promoter revertant mutation inhibited the expression of the Bcl-2 protein. The expression of the antiapoptotic protein Bcl-2 was downregulated, resulting in the loss of its inhibitory function against intrinsic apoptosis (Figure 6D,H and Appendix A). The expression of the pro-apoptotic proteins AIF, cytochrome C, and activated caspase 9 (cleaved caspase 9) was upregulated (Figure 6D,F,G,I,M,N and Appendix A). Expression of the mitochondrial proteins ANT and VDAC1 and the porin promoting protein Bax was upregulated (Figure 6E,F,J–L and Appendix A). VDAC1 and ANT are channel proteins in the mitochondrial membrane. Bax opens the mitochondrial membrane channel protein VDAC1 by binding to it or causing it to form a larger channel. Meanwhile, when Bcl-2 was inhibited, ANT was activated. Then, the opening of VDAC1 and ANT channels led to the release of the proapoptotic proteins AIF and cytochrome C into the cytoplasm. Subsequently, the release of cytochrome C induced the activation of caspase 9, which caused DNA damage and apoptosis (Figure 6O). These data suggested that the TERT promoter revertant mutation induced activation of the intrinsic apoptotic pathway by deregulating TERT inhibition of apoptosis. This ultimately led to growth inhibition of melanoma both in vitro and in vivo. 

## 4. Discussion

Cancer develops from oncogenic and tumor suppressor mutations; they initiate tumor progression and define cancers originating from accumulated genetic/epigenetic mutations, highlighting the role of gene therapy in cancer [35,52]. TERT overexpression and telomere activation have been detected in up to 90% of human malignancies [53,54]. Different activation strategies of telomerase in human malignancies have been shown to protect telomeres [52]. In tumor cells, mutant TERT promotes cell proliferation and invasion by inhibiting cell apoptosis [21]. 

It was previously reported that ETS1 and the GABPA/GABPB1 complex were bound to the ETS binding motif to activate telomerase [15,44,45,55]. The TERT promoter revertant mutation abolished the binding motif generated by mutant TERT, resulting in the inhibition of ETS binding to TERT [11]. On the basis of this, we found the TERT promoter revertant mutation downregulated the protein expression of ETS family members, including ETS1 (Figure 1E), GABPA (Figure 1F), and GABPB1 (Figure 1G). The reduction in ETS binding motifs, downregulation of ETS1, and reduction in the GABPA/GABPB1 complex also directly downregulated TERT transcription and protein expression (Figure 1D). Sanger sequencing of TERT and GABPA showed that the TERT promoter and GABPA were not mutated (Appendix A). The telomerase activity of A375^−146T/C^ and A375^−146C/C^ was also significantly reduced (Figure 1C). At the same time, the telomere length of A375^−146C/C^ was shortened (Figure 2B). Cell proliferation, migration and invasion of A375^−146T/C^ and A375^−146C/C^ cells were inhibited, and specifically, A375^−146C/C^ had almost no detectable migration or invasion (Figure 2C,D and Figure 4A). These results indicated that the TERT promoter revertant mutation abrogated the promotion of cell proliferation and invasion by mutant TERT. Next, cell-driven xenografts showed that A375^−146C/C^ could only form very small tumors, or no tumor formed at all, in nude mice (Figure 4B–E). HE staining of sections of xenograft tumors showed that A375^−146C/C^ formed tumors with a loose structure, low cell proliferation capacity, a lack of vascularity, and cavity-like structures inside (Figure 4F). Altogether, these results indicated that A375^−146C/C^ cells showed a very poor growth rates, both in vitro and in vivo. 

Intact cell structure and complete function are essential for mammalian cells. We observed that the morphology of A375^−146T/C^ and A375^−146C/C^ cells changed compared with A375^−146T/T^ cells, including cellular shrinkage, condensation margination of the chromatin and ruffling of the plasma membrane, suggesting that apoptosis occurred within these cells [45] (Figure 2E). The transcription and protein expression of β-Catenin and c-MYC, and the protein expression of PTEN were not significantly changed (Figure 5A,B). These results suggested that the TERT promoter revertant mutations may have no effect on Wnt/β-Catenin, and the TERT promoter revertant mutations may inhibit melanoma cell growth through other pathways.

The results of γ-H2AX immunofluorescence assay of A375-146T/T A375-146T/C and A375-146C/C showed that DNA damage is observed in cells with TERT revertant mutations (Figure 2F). By flow analysis, we again demonstrated that apoptosis occurred in cells with TERT revertant mutations. (Figure 6A–C). Then, we examined the expression of apoptosis-related proteins. We found that the inhibition of TERT in A375^−146T/C^ and A375^−146C/C^ cells induced a downregulation of Bcl-2 protein expression (Figure 6D), which led to an upregulation of pro-apoptotic proteins Bax, ANT, VDAC1, and AIF (Figure 6D–F), resulting in the release of large amounts of cytochrome C from the mitochondrial matrix into the cytoplasm, ultimately inducing intrinsic apoptosis through activated caspase 9 (Figure 6G). Herein, unlike the classical TERT pathway that affects cell growth and survival (TERT → telomerase → TEL → senescence) [11,29,32], our results showed that TERT promoter revertant mutation in melanoma did not induce cellular senescence but abrogated the inhibitory effect of mutant TERT on apoptosis (TERT → telomerase →TEL → Bcl-2 → apoptosis). These results indicated that TERT promoter revertant mutation not only inhibited melanoma growth but also caused to the occurrence of apoptosis. Similar results were observed in the xenografts. These findings indicated that TERT promoter revertant mutations caused melanoma growth inhibition by inducing apoptosis. 

It has been reported that inhibitors targeting TERT may induce apoptosis through DNA damage [45,56]. The pathway through which TERT reverts mutation induced apoptosis remains unclear. We found that Bcl-2 protein was high expressed in A375^−146T/T^ cells (high TERT expression) (Figure 6D), whereas the TERT revertant mutation downregulated TERT expression resulted in downregulation of Bcl-2 protein expression. These results implied that mutant TERT inhibits apoptosis in cancer cells, possibly by downregulating Bcl-2. 

In a previous study, reverting TERT promoter mutation has been reported to inhibit telomerase activity and growth of cancer cells [11]. Knockout of the TERT coding region was also found to inhibit the growth of cancer cells [29]. Developing and applying TERT as a therapeutic target in cancer have become popular research topics. The previously reported CjABE system was used to revert TERT -124 C>T, but its efficiency was low, and off-target effects were detected [11]. TERT exons have been knocked out using a dual sgRNA strategy and have been also shown to inhibit cell proliferation [29]. Due to the important role of TERT in somatic cells, off-target effects and TERT knockout may lead to DNA damage or some diseases. Thus, the CjABE system and TERT knockout may not be safe therapeutic options. With the intensive study of ABEs, most oncogene and anti-oncogene mutations can be modified [11,31,34]. NG-ABEmax had a high targeting effect, and was used to establish a precise point mutation in rabbits [39,57]. Here, we generated a TERT promoter revertant mutant using NG-ABEmax in A375^−146T/T^ cells. As expected, we found that the modification process was accurate and efficient, with no off-target mutations in potential off-target sites of the mutant cells (Figure 3B). Reverting TERT promoter mutation by NG-ABEmax may be a safe and effective potential cancer gene therapy targeting the TERT promoter. 

Furthermore, the encouraging results of recent work warrant follow-up studies in this area. Recently, various methods for transferring DNA and RNA into cells have been developed, such as lipofection, adeno-associated virus vector (AAV), nanoparticle delivery, microinjection, and acoustofluidic sonoporation [58]. These newly emerging proposed methods possibly solve the limitations of in vivo delivery. Meanwhile, our studies not only validated that the TERT promoter revertant mutations inhibit melanoma growth both in vitro and in vivo, but also elucidated for the first time the relationship between TERT promoter revertant mutation and apoptosis. We infer that that TERT -146 may be one of the causal mutations that affects melanoma growth, migration and invasion, and apoptosis. Recently, a preclinical study packaged CjABE into AAV for the treatment of TERT-124 C>T malignant glioma [11]. Although NG-ABEmax is highly efficient and without off-target mutations being detected, the SpCas9-NG is too large to be packaged into AAV for gene therapy. It is expected that smaller, highly efficient, highly targeted, and safe ABEs will be developed in the near future which are promising for the treatment of diseases by reverting causal mutations in vivo. Our analysis provided a preliminary validation for targeting revertant mutations of tumor-driven genetic mutations.

## 5. Conclusions

TERT promoter revertant mutation inhibits melanoma cell growth both in vitro and in vivo and promotes apoptosis. Based on these results, we infer that TERT -146 may be one of the causal mutations during the development of cancer. Our studies provided a preliminary confirmation for tumor therapy by reverting mutations.

## Figures and Tables

**Figure 1 biology-11-00141-f001:**
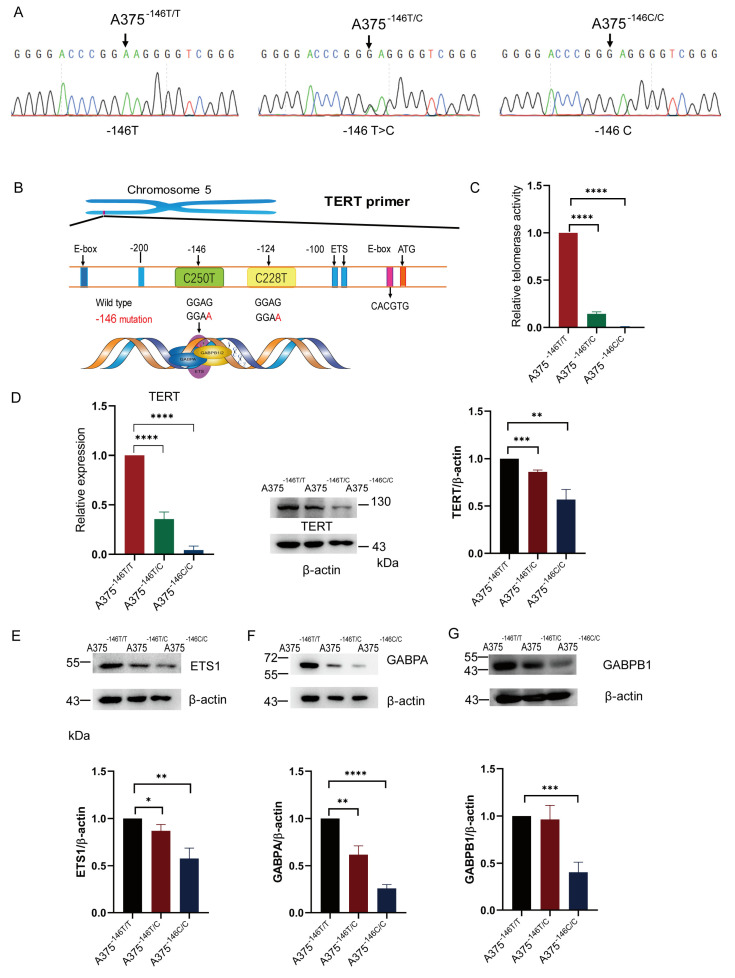
TERT -146 T>C inhibits TERT expression. (**A**) Sanger sequencing of A375^−146T/T^, A375^−146T/C^, and A375^−146C/C^. Cells were transfected with a plasmid expressing sgRNA and SpCas9-NG. The DNA covering −146 C>T in the TERT promoter was analyzed by Sanger sequencing. The arrows (upper panel) indicate mutations (lower panel). A375^−146T/T^, wildtype. A375^−146T/C^, heterozygous cells. A375^−146C/C^, homozygous cells; (**B**) schematic diagram of telomerase activation; (**C**) telomerase activity assay. Determination of telomerase activity by using the Real-time Q-TRAP; (**D**) TERT transcription and expression in A375^−146T/T^ and cells with TERT promoter revertant mutations; (**E**) Western blot and gray value analyses of ETS1 protein expression in A375^−146T/T^, A375^−146T/C^, and A375^−146C/C^; (**F**) Western blot and gray value analyses of GABPA protein expression in A375^−146T/T^, A375^−146T/C^, and A375^−146C/C^; (**G**) Western blot and gray value analyses of GABPB1 protein expression in A375^−146T/T^, A375^−146T/C^, and A375^−146C/C^. All results were obtained from the mean of at least three independent experiments. * *p* < 0.05, ** *p* < 0.01, *** *p* < 0.001, **** *p* < 0.0001, two-tailed unpaired *t*-test.

**Figure 2 biology-11-00141-f002:**
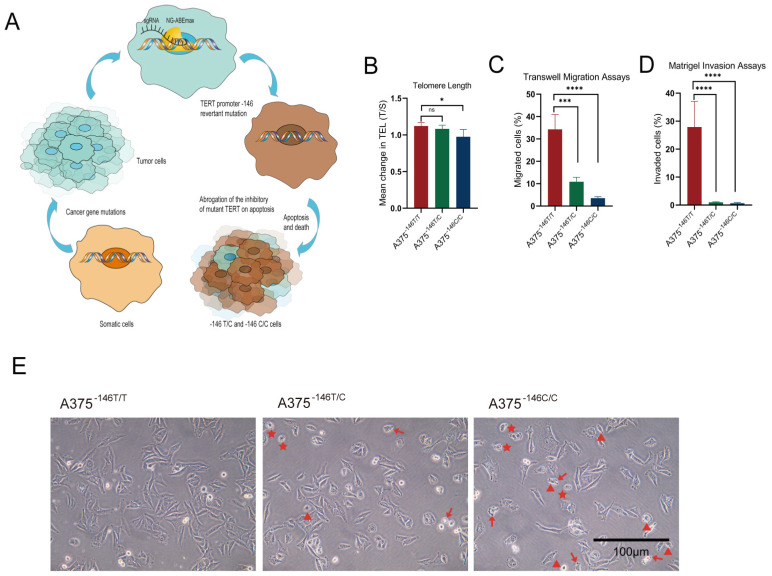
TERT -146 T>C reduces telomere length, migration and invasion of A375^−146T/C^ and A375^−146C/C^. (**A**) Schematic diagram of somatic cell carcinogenesis and revertant mutation; (**B**) telomere length measurements. DNA extraction from cells for Q-PCR. Mean change in telomere length (T/S) of A375^−146T/C^ and A375^−146C/C^. Reference cell line: HepG2; (**C**) cell migration assay. Migration and invasion tests measured by Transwell; (**D**) cell invasion assay; (**C**,**D**) the vertical coordinate indicates the percentage of migrated or invaded cells. Student’s *t*-test. * *p* < 0.05, *** *p* < 0.001, **** *p* < 0.0001; (**E**) changes in the morphology of A375^−146T/C^ and A375^−146C/C^ compared to A375^−146T/T^ (triangles: cellular shrinkage. pentagons: condensation and margination of chromatin. Arrows: ruffling of the plasma membrane) (original magnifications ×20. Scale bar, 100 µm); (**F**) DNA damage assay. Cells were cultured in 12-well plates and DNA damage was detected using γ-H2AX immunofluorescence assay. γ-H2AX immunofluorescence assay of A375-146T/T, A375-146T/C, and A375-146C/C. The green fluorescence indicates the amount of intracellular γ-H2AX, blue indicates DAPI staining. All results were obtained from the mean of at least three independent experiments.

**Figure 3 biology-11-00141-f003:**
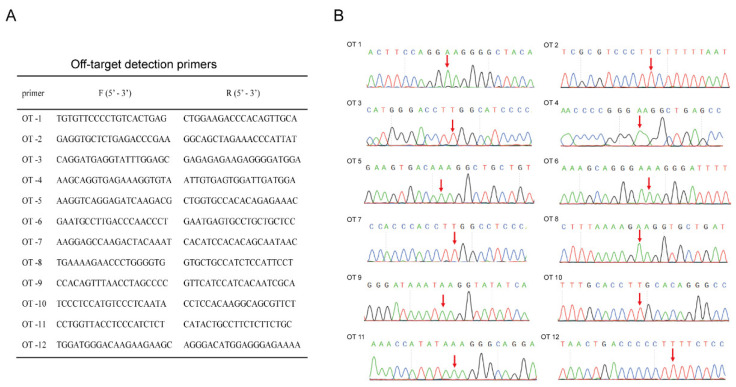
Off-target analysis of the cells with TERT promoter revertant mutations. (**A**) Primers for off-target analysis of the TERT promoter revertant mutation cells; (**B**) Sanger sequencing results of potential off-target sites of A375^−146C/C^. Potential mutation locations are marked with red arrows. OT: off-target.

**Figure 4 biology-11-00141-f004:**
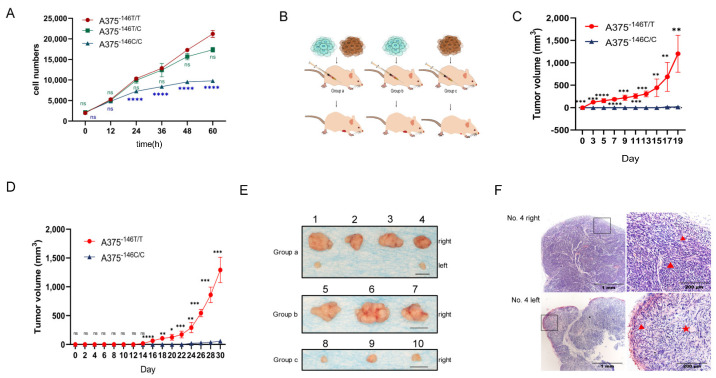
TERT -146 T>C inhibits melanoma growth. (**A**) Growth of A375^−146T/T^ and A375^−146T/C^ and A375^−146C/C^ in vitro. These cells were plated and counted at the indicated time points; (**B**) cell-derived xenografts of A375^−146T/T^ or A375^−146C/C^ cells in nude mice. Nude mice were divided into three groups, and A375^−146C/C^ or A375^−146T/T^ was injected into the anterior ventral side of each nude mouse as described schematically. Group a, n = 4; right: A375^−146T/T^, left: A375^−146C/C^. Group b and c, n = 6; group b right: A375^−146T/T^. Group c right, A375^−146C/C^ cells; (**C**) growth of A375^−146T/T^ and A375^−146C/C^ in nude mice of group a. After the third day of cell-derived xenografts in group a nude mice, the bilateral tumor size was measured at the indicated time points until the end of 19 days; (**D**) growth of A375^−146T/T^ or A375^−146C/C^ in nude mice of groups b and c. Tumor size was measured at the indicated time points until the end of 30 days; (**E**) typical tumors corresponding to 19 days and 30 days, respectively (scale bar, 1 cm); (Group a nude mice: No. 1, No. 2, No. 3, and No. 4; group b: No. 5, No. 6, and No. 7; group c: No. 8, No. 9, and No. 10); (**F**) hematoxylin-eosin staining of group a, No. 4. (Original magnifications of ×4 and ×20. Scale bar, 1 mm/200 µm); All results were obtained from the mean of at least three independent experiments. *t*-test, * *p* < 0.05, ** *p* < 0.01, *** *p* < 0.001, **** *p* < 0.0001.

**Figure 5 biology-11-00141-f005:**
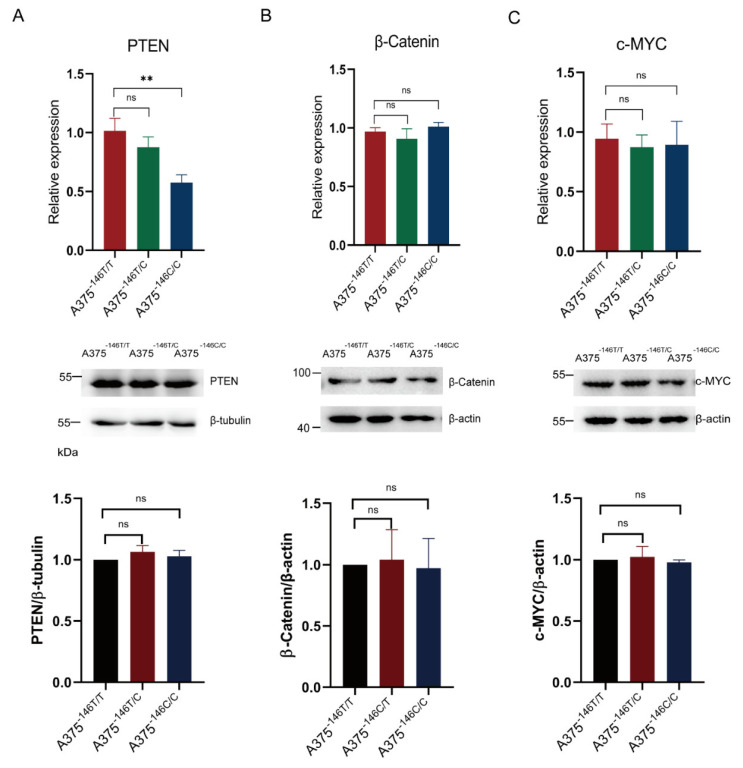
The TERT promoter revertant mutation and Wnt/β-Catenin signaling. (**A**) PTEN transcription expression and gray value analysis in A375^−146T/T^ and cells with TERT promoter revertant mutations; (**B**) β-Catenin transcription expression and gray value analysis in A375^−146T/T^ and cells with TERT promoter revertant mutations; (**C**) c-MYC transcription expression and gray value analysis in A375^−146T/T^ and cells with TERT promoter revertant mutations. Student’s *t*-test. ** *p* < 0.01.

**Figure 6 biology-11-00141-f006:**
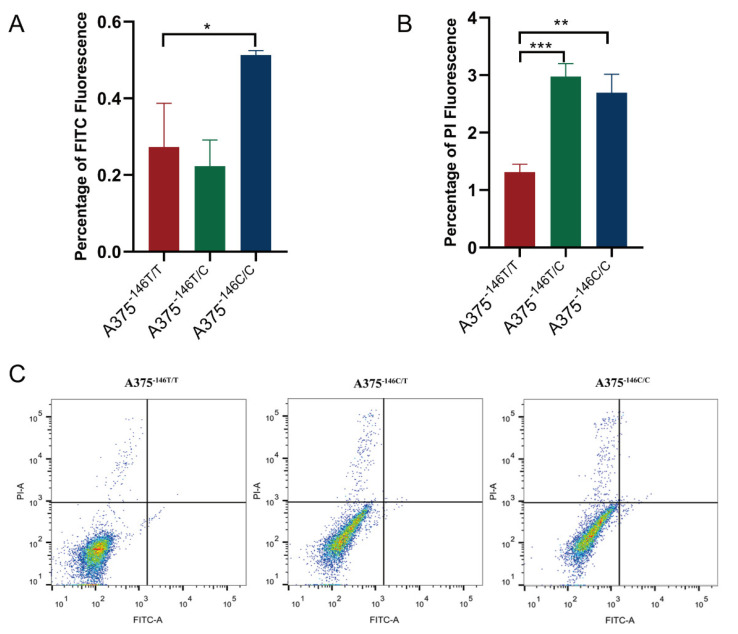
Abrogation of the inhibitory effect of mutant TERT on apoptosis. (**A**–**C**) FACs analysis with Annexin V-FITC/PI; (**A**,**C**) the early apoptosis detection; (**B**,**C**) the late stage apoptosis detection; (**D**,**H**,**I**) Western blot and gray value analysis of Bcl-2 and Cyt c protein expression in A375^−146T/T^, A375^−146T/C^, and A375^−146C/C^; (**E**,**J**) Western blot and gray value analysis of ANT expression in A375^−146T/T^, A375^−146T/C^, and A375^−146C/C^; (**F**,**K**–**M**) Western blot analysis of Bax, VDAC1, and AIF expression in A375^−146T/T^, A375^−146T/C^, and A375^−146C/C^; (**G**,**N**) Western blot analysis of caspase 9 expression in A375^−146T/T^, A375^−146T/C^, and A375^−146C/C^; (**O**) schematic diagram of intrinsic apoptosis induced by TERT revertant mutation. AIF: apoptosis-inducing factor; VDAC1: voltage-dependent anion channel 1; ANT: adenine nucleotide translocator; Bax: Bcl-2-associated X protein. * *p* < 0.05, ** *p* < 0.01, *** *p* < 0.001, **** *p* < 0.0001, *t*-test.

## Data Availability

Information on melanoma incidence and death rates displayed in this study can be found at http://www.healthdata.org/ (accessed on 1 September 2021).

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
