# Peer review of "TERT Promoter Revertant Mutation Inhibits Melanoma Growth through Intrinsic Apoptosis"

_biology, 2022, doi:10.3390/biology11010141_

Round 1
Reviewer 1 Report
General observations to the authors:
This is an interesting article which showed that reverting of mutations in the TERT promoter could be potential therapeutic strategy for melanoma. However, several issues in manuscript need to be fix.
Q1.- For better clarity, raw Western blotting data should be provided with well labeling of ladder size and indicating the site of cropping in full membrane. Few membranes are so closely cropped (Fig1 GABPA, TERT, Fig.6 Bax, VDAC1, ANT, AIF).
Q2.- To support TERT promoter revertant mutation induces apoptosis, authors should include FACs analysis with Annexin V in Fig 6.
Q3. All Western blotting band in figures should be quantified.
Q4. Western blotting of key proteins should be analyzed in lysate of tumor obtained from in vivo stody in figure 4.
Q5. It will be supportive if authors could determine TERT promoter activity in Figure 1.
Q6. I would suggest authors to use appropriate statistical analysis. One‑way ANOVA followed by post hoc test would be recommended.
Author Response
Dear editor and reviewers:
Thank you very much for your comments concerning our manuscript entitled “TERT promoter revertant mutation inhibits melanoma growth through intrinsic apoptosis”. Those comments are all valuable and very helpful for revising and improving our paper, as well as the important guiding significance to our researches. We have revised the manuscript accordingly and a detailed response to the reviewers’ comments has been provided below.
The revised manuscript is attached
To Reviewer #1:
Q1.- For better clarity, raw Western blotting data should be provided with well labeling of ladder size and indicating the site of cropping in full membrane. Few membranes are so closely cropped (Fig1 GABPA, TERT, Fig.6 Bax, VDAC1, ANT, AIF).
Response:
Thanks a lot for your patient and careful reading of our manuscript. We realized that the membranes were cropped too tightly, so we re-performed the western blot (Fig1 GABPA, TERT, Fig.6 Bax, VDAC1, ANT, AIF and so on).
Q2.- To support TERT promoter revertant mutation induces apoptosis, authors should include FACs analysis with Annexin V in Fig 6.
Response:
Thank you for pointing this out. I apologize for our lack of thoughtfulness. FACs analysis with Annexin V can be found in Fig 6A, B and C. We found that compared to A375-146T/T, early apoptosis occurs in A375-146C/C cells, while both early and late apoptosis occur in A375-146T/C and A375-146C/C cells
Q3. All Western blotting band in figures should be quantified.
Response:
Thank you very much for your kindly comments and we are very sorry for the negligence. We have supplemented some western blot results and quantified all the western blotting band in figures. Please view in the article.
Q4. Western blotting of key proteins should be analyzed in lysate of tumor obtained from in vivo stody in figure 4.
Response:
Thank you very much for your kindly reminder and we are very sorry for the negligence. We have seriously thought about this question. The reasons why we did not use tumor lysate for western blotting are as follows:
- There are multiple unpredictable effects in nude mice that could affect the critical pathway leading us to implausible results.
- In the process of culturing cells in vitro, basically no factors other than our previous gene editing (revertant mutation-146T>C) affect the growth state of the cells and do not affect the apoptotic pathway. Only then can we get a relatively accurate result.
- Some of these tumors are too small even for doing HE staining is a bit insufficient. We only had to use them for HE staining. Meanwhile, our HE staining results also showed well that apoptosis was indeed present in the tumor.
Q5. It will be supportive if authors could determine TERT promoter activity in Figure 1.
Response:
Thank you very much for your kindly reminder and we are very sorry for the negligence. After reviewing the extensive similar literature[1-3], we found that the majority of authors chose to assay telomerase activity rather than TERT promoter activity, Also, telomerase activity can indirectly reflect the activity of TERT promoter. We assayed the telomerase activity of A375-146T/T A375-146T/C and A375-146C/C as previously described[4]. The results of telomerase assay are shown in Fig 1C.
Q6. I would suggest authors to use appropriate statistical analysis. One‑way ANOVA followed by post hoc test would be recommended.
Response:
Thank you for pointing this out. We would like to express our sincere apologies for not using the appropriate analysis method before. In our statistics, the control is A375-146T/T(WT). A375-146T/C (heterozygote) and A375-146C/C (homozygote) only need to be compared with the control and not with each other, so we should choose the t-test. All inappropriate analysis methods have been corrected. Once again, we apologize to you.
- Werner, C.M.; Hecksteden, A.; Morsch, A.; Zundler, J.; Wegmann, M.; Kratzsch, J.; Thiery, J.; Hohl, M.; Bittenbring, J.T.; Neumann, F.; et al. Differential effects of endurance, interval, and resistance training on telomerase activity and telomere length in a randomized, controlled study. European heart journal 2019, 40, 34-46, doi:10.1093/eurheartj/ehy585.
- Konieczna, N.; Romaniuk-Drapała, A.; Lisiak, N.; Totoń, E.; Paszel-Jaworska, A.; Kaczmarek, M.; Rubiś, B. Telomerase Inhibitor TMPyP4 Alters Adhesion and Migration of Breast-Cancer Cells MCF7 and MDA-MB-231. International journal of molecular sciences 2019, 20, doi:10.3390/ijms20112670.
- Wang, F.; Cheng, Y.; Zhang, C.; Chang, G.; Geng, X. A novel antisense oligonucleotide anchored on the intronic splicing enhancer of hTERT pre-mRNA inhibits telomerase activity and induces apoptosis in glioma cells. Journal of neuro-oncology 2019, 143, 57-68, doi:10.1007/s11060-019-03150-x.
- Herbert, B.S.; Hochreiter, A.E.; Wright, W.E.; Shay, J.W. Nonradioactive detection of telomerase activity using the telomeric repeat amplification protocol. Nature protocols 2006, 1, 1583-1590, doi:10.1038/nprot.2006.239.

Reviewer 2 Report
This is an interesting manuscript by Wang et al that describes the role of -146 C>T mutation at TERT promoter region on tumor growth and survival. Although, this manuscript is relevant and includes novel finding that mutation of TERT at -146 regulates the expression of TERT, ETS, GABPA and GABPB1, it falls short of some key data to make such strong conclusions. Here are some of the major comments.
a. This Manuscript is highly descriptive with long unnecessary introduction that needs to be focussed and shortened. The result section are also highly descriptive and not focussed to the point of the experimental design and observed result.
b. Figure 1 shows that the authors made successful mutation at -146 region. However, the authors should evaluate/confirm whether this experimental method affected other regions in the TERT promoter (such as -124 and start site) that was previously shown to regulate TERT expression.
c. The authors refer to non-experimental figures in the manuscript as a data point which should be strictly avoided. For example, they refer to Fig 1b which is a schematic representation and state that -146 mutation reduced ETS binding. similarly, they refer fig 2a (which is also a schematic) and state that majority of mixed cells after revert mutation underwent apoptosis.
d. The authors show that mutation of -146 C>T reduced TERT expression and surprisingly also reduced ETS, GABPA, GABPB1 expression. Given that ETS, GABPA, GABPB1 proteins regulate TERT expression (among many other) and since it is not known if TERT can regulate the expression of these proteins, the authors should evaluate the mutations status of these genes (an off target effect or -146 mutation) and further discuss how TERT regulate ETS, GABPA, GABPB1 levels.
e. Given the data on TERT levels regulating cellular apoptosis. The authors should measure if loss of TERT expression induced DNA damage (measure y-H2AX) that can explain the increased apoptosis and revise the schematic figure 6F.
f. The authors conducted their experiments on one cell line. However, since different cell lines respond differently to mutations in -124 C>T, experiments in other cell lines especially measuring the levels of TERT upon -146 C>T mutation is highly recommended.
Minor comments.
a. The authors should revise their figure legends to include the experimental design (such as time points of conducting and collection).
b. Should revise their description of experimental design of figure 4 from line 312 to 319, as it is incomprehensible.
c. Figure 1A and 2E should be increased in size as the fine details are un-readable. Especially Fig 2E should include zoomed in images of few cells to clearly show the differences in morphology.
Round 2
Reviewer 2 Report
This reviewer is very satisfied within the changes in this revised manuscript. However, some minor corrections on figure placement still remain, which needs to be addressed.
1. As worded by the authors, Fig 1B can be placed after the citation [18] in line 256 but not after "-146 T>C decreased ETS binding motif". As this sentence indicates an experimental result but not a schematic drawing.
2. Similarly, as written by the authors in line 289 " we found that most of the mixed cells after the reversion mutation underwent apoptosis" (Fig. 2A). This sentence suggests an experimental result of changes in apoptosis. The author should cite the right figure (that includes experimental result) indicating changes in apoptosis and can follow up with this schematic.
3. Finally, Figure 6O should also incorporate the DNA damage caused by -146 C/C mutation.
4. On a side not the Figure 6N (Caspase 9) shows same labels (-146T/C).
Author Response
- As worded by the authors, Fig 1B can be placed after the citation [18] in line 256 but not after "-146 T>C decreased ETS binding motif". As this sentence indicates an experimental result but not a schematic drawing.
Response:
Thanks for your kindly advices. We have put Figure 1B in the right place as you suggested, please check in line 256 in the manuscript (Marked in green). We updated the reference resulting in [18] to [17]
- Similarly, as written by the authors in line 289 " we found that most of the mixed cells after the reversion mutation underwent apoptosis" (Fig. 2A). This sentence suggests an experimental result of changes in apoptosis. The author should cite the right figure (that includes experimental result) indicating changes in apoptosis and can follow up with this schematic.
Response:
Thanks for your kindly advices. We have put Figure 2A in the right place as you suggested, please check in line 289-291 in the manuscript (Marked in green).
- Finally, Figure 6O should also incorporate the DNA damage caused by -146 C/C mutation.
Response:
Thanks a lot for your patient and careful reading of our manuscript. We added DNA damage (DNA fragmentation) in Figure 6O.
- On a side not the Figure 6N (Caspase 9) shows same labels (-146T/C).
Response:
Thank you very much for your kindly advices and we are very sorry to confuse you. We wrote the wrong title for the Figure 6N vertical coordinate, the correct title should be: cleaved-casp 9/total-casp 9, and we have modified it.